# Number of Retrieved Lymph Nodes during Esophagectomy Affects the Outcome of Stage III Esophageal Cancer in Patients Having Had Pre-Operative Chemo-Radiation Therapy

Wei Ho [1,†], Shau-Hsuan Li [2,†], Shih-Ting Liang [3], Yu Chen [1], Li-Chun Chen [1], Yen-Hao Chen [2], Hung-I Lu [1]⬤ and Chien-Ming Lo [1,*]⬤

[1] Department of Thoracic & Cardiovascular Surgery, Kaohsiung Chang Gung Memorial Hospital, Chang Gung University College of Medicine, Kaohsiung 833, Taiwan; kobewei5516@yahoo.com.tw (W.H.); allen000001@gmail.com (Y.C.); lisachen143@cgmh.org.tw (L.-C.C.); luhungi@cgmh.org.tw (H.-I.L.)

[2] Department of Hematology-Oncology, Kaohsiung Chang Gung Memorial Hospital, Chang Gung University College of Medicine, Kaohsiung 833, Taiwan; lee.a0928@msa.hinet.net (S.-H.L.); alex2999@cgmh.org.tw (Y.-H.C.)

[3] Department of Nurse, Kaohsiung Chang Gung Memorial Hospital, Chang Gung University College of Medicine, Kaohsiung 833, Taiwan; s3471020@cgmh.org.tw

[*] Correspondence: t123207424@cgmh.org.tw or johncml9487@gmail.com or pichupikachu69@hotmail.com; Tel.: +886-7-7317123 (ext. 8303); Fax: +886-7-7322402

[†] These authors contributed equally to this work.

**Abstract:** *Background*: Lymphadenectomy plays a crucial role in the surgical management of early-stage esophageal cancer. However, few studies have examined lymphadenectomy outcomes in advanced stages, particularly in patients who initially underwent concurrent chemoradiation therapy. This retrospective study investigates the effect of lymphadenectomy in patients diagnosed with AJCC 8th-edition clinical stage III esophageal squamous cell carcinoma who received concurrent preoperative chemoradiation. *Methods*: Data from 1994 to 2023 were retrieved from our retrospective database. All patients underwent a uniform evaluation and treatment protocol, including preoperative concurrent chemoradiation therapy comprising cisplatin and 5-fluorouracil, followed by esophagectomy. The analysis encompassed clinical T and N stages, tumor location, tumor grade, pathological T and N stages, pathological stage, and the extent of lymph node dissection. Overall survival, "Free-To-Recurrence", and disease-free survival were assessed via Kaplan–Meier survival curves and the Cox regression model for multivariate analysis. *Results*: The dataset was stratified into two groups according to extent of lymph node dissection, with one group having <15 dissected nodes and the other having ≥15 dissected nodes. The group with <15 nodes exhibited a shorter "Free-To-Recurrence", worse disease-free survival, and lower overall survival. In multiple-variate analysis (Cox regression model), the number of dissected lymph nodes emerged as a significant factor influencing overall survival and freedom from recurrence. *Conclusions*: The quantity of lymphadenectomy is a crucial determinant for patients with AJCC 8th-edition clinical stage III esophageal squamous cell carcinoma receiving preoperative concurrent chemoradiation.

**Keywords:** esophageal cancer; chemoradiation; lymph node dissection; esophagectomy; esophageal squamous cell carcinoma; clinical stage III

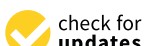



## 1. Introduction

Esophageal cancer, the tenth most commonly diagnosed cancer worldwide in 2020, is a serious and often fatal condition [1]. Most patients are only diagnosed at advanced stages, as they are usually asymptomatic. According to the CROSS trial, one of the recommended treatment options for locally advanced esophageal cancer is concurrent chemoradiotherapy followed by esophagectomy [2], which was also proven beneficial using a large randomized

trial [3] and meta-analysis [4]. Surgery has been shown to improve survival rates, even after induction therapy, and has been set as a treatment guideline [5].

Radical lymphadenectomy is crucial for accurate staging and curative treatment [6]. The importance of radical lymphadenectomy after preoperative concurrent chemoradiation remains controversial. Some studies have shown its significance [7–9], but another series has cast doubt on this [10]. Despite some studies suggesting that induction therapy can reduce the necessity of lymphadenectomy, the appropriate amount of lymph node dissection required remains debatable.

Apart from the debate on the effect of lymphadenectomy on esophageal cancer treatment, specific stage treatment lymphadenectomy results remain lacking and controversial. We hypothesized that the number of dissected lymph nodes is related to overall survival and recurrence in patients with AJCC 8th-edition clinical stage III esophageal squamous cell carcinoma. In this retrospective study, we aimed to clarify the benefits of radical lymphadenectomy, the number of pathologic lymph nodes, and their impact on survival and recurrence rates.

## 2. Materials and Methods

### 2.1. Patient Population

Patients with AJCC 8th-edition clinical stage III esophageal cancer, who underwent preoperative concurrent chemoradiation followed by esophagectomy between 1 July 1994 and 30 June 2023 at the Kaohsiung Branch of Chang Gung Memorial Hospital, were reviewed retrospectively. Excluded patients included those without clear medical records, those undergoing treatment at other hospitals, those undergoing radiotherapy alone, those undergoing treatment protocols other than concurrent chemoradiation therapy followed by esophagectomy, those lost to follow-up, those undergoing conservative or hospice care after diagnosis, and those failing to complete the treatment course. All patients underwent a similar staging protocol, including computed tomography of the chest with and without contrast, endoscopic ultrasound, and/or whole-body Positron Emission Tomography after biopsy-confirmed esophageal squamous cell carcinoma. Patients with other pathologies, such as small cell carcinoma, were also excluded. A clinical stage depends on these staging tools. Where there are conflicting results, multidisciplinary teams will discuss it with gastrointestinal physicians, radiology physicians, surgeons, and oncologists to confirm the clinical stage. The tumor stage was determined according to the eighth edition of the American Joint Committee on Cancer (AJCC) staging system. We analyzed these patients into two groups, of which one is retrieved lymph nodes numbers less than 15, while another group contains numbers equal or more than 15. We separated the two groups according to the study from J. C. Yeung et al. [11].

This study was approved by the Institutional Review Board of the Kaohsiung Chuang Gung Memorial Hospital (IRB No: 202400100B0).

### 2.2. Treatment Plan

After confirming the study and performing tumor staging, all patients received concurrent chemoradiation therapy. Chemotherapy included 2 cycles of cisplatin- (75 mg/m$^2$; 4 h drip) and 5-fluorouracil-based (1000 mg/m$^2$; continuous infusion) treatment, administered on days 1–4, every 4 weeks. Radiotherapy was administered at three different doses (3600, 4140, and 5000 cGy) in continued 5-day fractions per week. Three-dimensional conformal radiotherapy (CRT) via a four-field technique or intensity-modulated radiotherapy (IMRT) with 6 or 10 MV photons was utilized. Gross tumors and lymph nodes on computed tomography and/or whole-body Positron Emission Tomography were defined as the gross target volume (GTV). The treatment area encompassed by the clinical target volume (CTV) included the esophagus, mediastinal lymph nodes, both sides of the neck, and lymph nodes above the clavicle. Expanding from the CTVs, the planning target volume (PTV) had an added margin of 0.5–1.0 cm in all directions. Within 3–4 weeks post-irradiation, a series of evaluations, including CT scans from the neck to the upper abdomen, endoscopic

examinations, and/or PET/CT scans, were conducted to assess treatment response. After these surveys, we conducted multiple multidisciplinary team meetings with the surgeon, to arrange esophagectomy if feasible. The actual surgical timing is around 6–8 weeks after chemoradiation.

Three surgeons performed esophagectomies using the McKeown procedure for all patients. Consistency was maintained across the operating room configurations, team compositions, and surgical instruments. All three surgeons performed routine lymph node dissection from the subcarinal, paraesophageal, bilateral recurrent laryngeal nerve area, celiac, and perigastric lymph nodes. Specimens obtained from esophagectomies were forwarded to the pathology laboratory for comprehensive assessments, encompassing the entire excised esophagus, as well as the thoracic and abdominal lymph nodes. Pathologists evaluated tumor characteristics, depth, lymphovascular invasion, resection margins, and staging in accordance with the American Joint Committee on Cancer eighth-edition guidelines.

### 2.3. Overall Survival, Free-to-Recurrence and Disease-Free Survival

Results between different lymph node amounts were compared using the three datasets. Overall survival was defined as the period from the date of the first diagnosis of esophageal cancer to the last contact date. If a patient died (regardless of cause), it was delimited as an event. Patients who survived were censored. We collected data on "Free-To-Recurrence", which is defined as the time from the date of curative surgery to the time of recurrence. If a patient died without recurrence, it was delimited as a censor. If a patient had recurrence, it was delimited as an event [12]. We collected data on "Disease-free Survival", which is defined as the period from the curative surgery to the time of recurrence. If a patient died (regardless of cause of death) or experienced recurrence, we delimited it as an event. Patients who survived without recurrence were censored. The difference between "Free-To-Recurrence" and "Disease-free survival" is "death". "Free-To-Recurrence" did not include death as an event, but "Disease-free survival" included death as an event.

### 2.4. Statistical Analysis

Data were analyzed using MedCalc® Statistical Software version 20 (MedCalc Software Ltd., Ostend, Belgium; https://www.medcalc.org; 26 August 2021). A $\chi^2$ test was employed to compare data across the two groups. Univariate survival analysis was conducted using the Kaplan–Meier method, and differences in survival rates were assessed using a log-rank test. Factors were sequentially integrated into a Cox regression model using "Enter" fashion to evaluate their respective prognostic significance. Two-sided tests of significance were applied for all analyses, with statistical significance denoted as $p < 0.05$. We selected the factors which illustrated in the "Overall Survival, Free-To-Recurrence and Disease-free survival". These factors included clinical T stage, clinical N stage, tumor location, tumor grade, and lymph node amounts. We excluded the pathology T and N stage which are highly associated with clinical T and N stages.

## 3. Results

### 3.1. Patient Characteristics

This study enrolled 91 patients, whose basic profiles are summarized in Table 1. The mean age, median age, and age range were 55.66, 55, and 36–76 years, respectively. All patients were pathologically diagnosed with squamous cell carcinoma. Only three patients were females (3.2%). Most patients were clinically diagnosed at the T3 stage (91.2%). Overall, 15 patients had tumors in the upper third of the esophagus (16.5%), 40 in the middle third (44.0%), and 36 in the lower third (39.5%). Fifty-three patients (58.2%) revealed moderate tumor differentiation. Twenty-six patients (28.6%) achieved pathologic complete response. The median number of retrieved lymph nodes was 19. Even though concurrent chemoradiotherapy was administered prior to operation, seven patients (7.7%)

progressed to T4b lesions, and the stage progressed to IVA. Twenty-eight patients (30.8%) could not obtain >15 lymph nodes under radical dissection.

**Table 1.** Clinicopathologic factors of 91 patients with 8th AJCC clinical stage 3 esophageal squamous cell carcinoma (AJCC, American Joint Committee on Cancer).

| Factors | No. of Patients (Percentage) |
|---|---|
| Age (years) (range: 36–76, mean: 55.66, median: 55) | |
| Gender | |
| Male | 88 (96.7%) |
| Female | 3 (3.3%) |
| Clinical T stage | |
| T1b | 4 (4.4%) |
| T2 | 4 (4.4%) |
| T3 | 83 (91.2%) |
| Clinical N stage | |
| N1 | 43 (47.3%) |
| N2 | 48 (52.7%) |
| Primary tumor location | |
| Upper | 15 (16.5%) |
| Middle | 40 (44.0%) |
| Lower | 36 (39.5%) |
| Pathological tumor grade | |
| 0 (Tis) | 26 (28.6%) |
| 1 | 2 (2.2%) |
| 2 | 53 (58.2%) |
| 3 | 10 (11.0%) |
| Pathologic 8th AJCC stage | |
| 0 | 27 (29.6%) |
| IA | 3 (3.3%) |
| IB | 9 (9.9%) |
| IIA | 9 (9.9%) |
| IIB | 18 (19.8%) |
| IIIA | 8 (8.8%) |
| IIIB | 8 (8.8%) |
| IVA | 9 (9.9%) |
| Pathologic T stage | |
| 0 | 29 (31.8%) |
| 1a | 2 (2.2%) |
| 1b | 14 (15.4%) |
| 2 | 12 (13.2%) |
| 3 | 25 (27.5%) |
| 4a | 2 (2.2%) |
| 4b | 7 (7.7%) |
| Pathologic N stage | |
| 0 | 68 (74.7%) |
| 1 | 19 (20.9%) |
| 2 | 3 (3.3%) |
| 3 | 1 (1.1%) |
| Pathologic lymph node amount | |
| <15 | 28 (30.8%) |
| ≥15 | 63 (69.2%) |

Patients were divided into two groups according to the number of pathological lymph nodes. We used 15 as the cut-off value and compared the basic data between the groups, as shown in Table 2. None of the parameters, including age, sex, clinical T stage, clinical N stage, tumor location, pathologic tumor grade, pathologic stage, pathologic T stage, and pathologic N stage, revealed significant differences between the two groups.

**Table 2.** Comparison features of esophageal squamous cell carcinoma patients with 8th AJCC pathologic lymph node amount <15 or ≥15.

| Parameters | | Lymph Nodes Amount | | |
|---|---|---|---|---|
| | | <15 (28) | ≥15 (63) | *p*-Value |
| Age (years) (Mean ± standard deviation) | | 55.6 ± 8.3 | 55.6 ± 8.5 | 0.99 |
| Gender | Male | 28 | 60 | 0.24 |
| | Female | 0 | 3 | |
| Clinical T stage | T1b | 0 | 4 | 0.29 |
| | T2 | 2 | 2 | |
| | T3 | 26 | 57 | |
| Clinical N stage | N1 | 11 | 32 | 0.31 |
| | N2 | 17 | 31 | |
| Primary tumor location | Upper | 4 | 11 | 0.67 |
| | Middle | 11 | 29 | |
| | Lower | 13 | 23 | |
| Pathologic tumor grade | 0 (Tis) | 7 | 19 | 0.36 |
| | 1 | 1 | 1 | |
| | 2 | 19 | 34 | |
| | 3 | 1 | 9 | |
| Pathologic 8th AJCC stage | 0 | 7 | 20 | 0.44 |
| | IA | 2 | 1 | |
| | IB | 3 | 6 | |
| | IIA | 3 | 6 | |
| | IIB | 3 | 15 | |
| | IIIA | 4 | 4 | |
| | IIIB | 4 | 4 | |
| | IVA | 2 | 7 | |
| Pathologic T stage | 0 | 7 | 22 | 0.51 |
| | 1a | 1 | 1 | |
| | 1b | 6 | 8 | |
| | 2 | 2 | 10 | |
| | 3 | 10 | 15 | |
| | 4a | 1 | 1 | |
| | 4b | 1 | 6 | |
| Pathologic N stage | 0 | 19 | 49 | 0.42 |
| | 1 | 7 | 12 | |
| | 2 | 2 | 1 | |
| | 3 | 0 | 1 | |

$\chi^2$ test or *t* test was utilized for statistical analysis.

### 3.2. Overall Survival, Free-to-Recurrence, and Disease-Free Survival

At the time of analysis, the median follow-up duration for the 38 survivors was 68.3 months (ranging from 6.7 to 168.9 months), while for the entire cohort of 91 patients it was 31.5 months (ranging from 5.6 to 168.9 months). The average follow-up period was 69.0 months for the survivors and 45.8 months for all patients. Correlations of overall survival, recurrence-free survival, disease-free survival, and parameters are summarized in Table 3. Pathological tumor grade, pathological stage, pathological T stage, and lymph node volume from surgery had significant impacts on overall survival. The *p*-values were 0.0473 for pathological tumor grade, 0.0034 for pathological stage, 0.0004 for pathological T stage, and 0.0165 for pathological lymph nodes. Kaplan–Meier survival curves across different pathologic lymph node groups are illustrated in Figure 1.

As for recurrence-free survival, pathologic stage, pathologic T stage, pathologic N stage, and pathologic lymph node amount contributed for a significant influence. The *p*-values were 0.0001 for the pathologic stage, <0.0001 for the pathologic T stage, 0.0081 for the pathologic N stage, and 0.0363 for the lymph node amount. Kaplan–Meier survival curves between the different pathologic lymph node groups are illustrated in Figure 2.

**Table 3.** Results of univariate analysis for overall and relapse-free survivals in 91 patients with 8th AJCC clinical stage III esophageal squamous cell carcinoma.

| Factors | No.p't | Overall Survival | | Time to Recurrence | | Disease-Free Survival | |
|---|---|---|---|---|---|---|---|
| | | 3-y OS (%) | p | 3-y TTR (%) | p | 3-y DFS (%) | p |
| Clinical T stage | | | | | | | |
| T1b | 4 | 75% | 0.96 | 67% | 0.53 | 50% | 0.51 |
| T2 | 4 | 50% | | 33% | | 25% | |
| T3 | 83 | 58% | | 57% | | 45% | |
| Clinical N stage | | | | | | | |
| N1 | 43 | 59% | 0.84 | 52% | 0.43 | 44% | 0.78 |
| N2 | 48 | 58% | | 60% | | 45% | |
| Primary tumor location | | | | | | | |
| Upper | 15 | 45% | 0.27 | 43% | 0.26 | 36% | 0.38 |
| Middle | 40 | 69% | | 70% | | 54% | |
| Lower | 36 | 51% | | 48% | | 37% | |
| Pathologic tumor grade | | | | | | | |
| 0 | 26 | 76% | 0.0473 * | 85% | 0.074 | 67% | 0.123 |
| 1 | 2 | 50% | | 100% | | 50% | |
| 2 | 53 | 52% | | 45% | | 33% | |
| 3 | 10 | 50% | | 56% | | 50% | |
| Pathologic AJCC stage | | | | | | | |
| 0 | 27 | 77% | 0.0034 * | 86% | 0.0001 * | 69% | 0.0076 * |
| IA | 3 | 67% | | 100% | | 67% | |
| IB | 9 | 65% | | 50% | | 42% | |
| IIA | 9 | 67% | | 78% | | 67% | |
| IIB | 18 | 66% | | 59% | | 31% | |
| IIIA | 8 | 45% | | 38% | | 38% | |
| IIIB | 8 | 29% | | 17% | | 15% | |
| IVA | 9 | 13% | | 0% | | 0% | |
| Pathologic T stage | | | | | | | |
| 0 | 29 | 79% | 0.0004 * | 87% | <0.0001 * | 71% | 0.0006 * |
| T1a | 2 | 50% | | 50% | | 50% | |
| T1b | 14 | 77% | | 61% | | 48% | |
| 2 | 12 | 75% | | 74% | | 50% | |
| 3 | 25 | 35% | | 36% | | 27% | |
| T4a | 2 | 50% | | 0% | | 0% | |
| T4b | 7 | 0% | | 0% | | 0% | |
| Pathologic N stage | | | | | | | |
| 0 | 68 | 63% | 0.12 | 67% | 0.0081 * | 52% | 0.146 |
| 1 | 19 | 51% | | 33% | | 26% | |
| 2 | 3 | 0% | | 0% | | 0% | |
| 3 | 1 | 0% | | 0% | | 0% | |
| Lymph node amount | | | | | | | |
| <15 | 28 | 46% | 0.0165 * | 40% | 0.0363 * | 29% | 0.0114 * |
| ≥15 | 63 | 64% | | 64% | | 52% | |

* statistically significant. OS, overall survival; AJCC, American Joint Committee on Cancer; TTR, Time to Recurrence; DFS, disease-free survival.

In our study, only the pathologic stage, the pathologic T stage, and the number of lymph nodes were related to disease-free survival. The *p*-values were 0.0076, 0.0006, and 0.0114, respectively. The Kaplan–Meier survival curve between the different pathologic lymph node groups is illustrated in Figure 3. The median number of retrieved lymph nodes was 19 and illustrated in Figure 4.

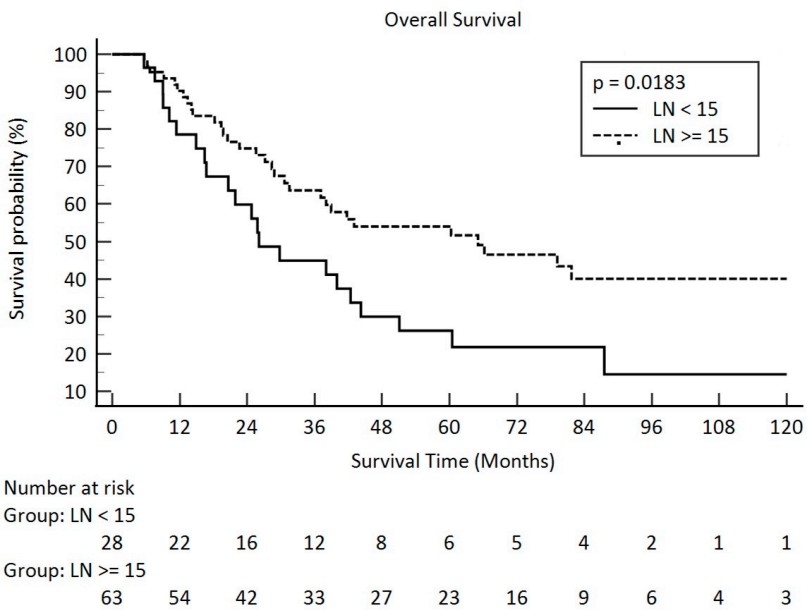

**Figure 1.** Overall survival in pathologic lymph node amount after esophagectomy for <15 and ≥15 in clinical stage III esophageal squamous cell carcinoma (solid line: <15; dot line: ≥15).

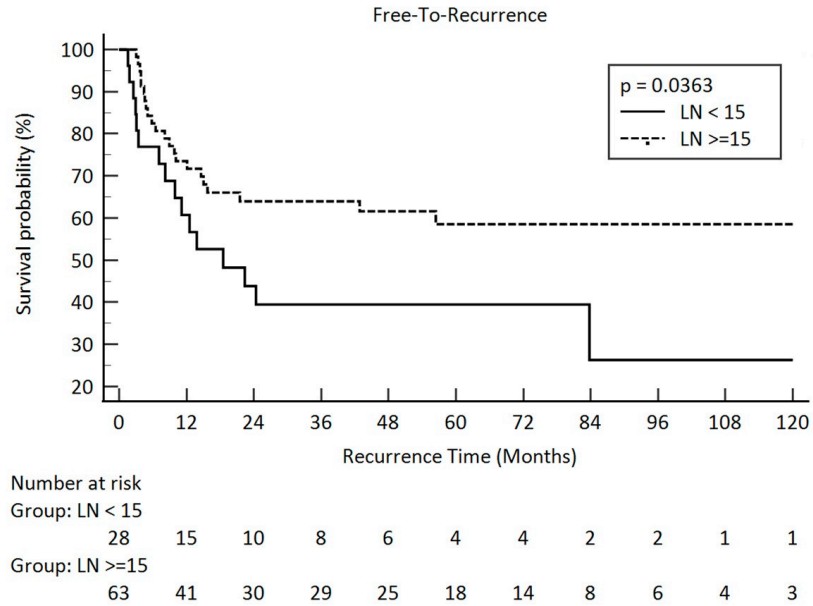

**Figure 2.** "Free-To-Recurrence" in pathologic lymph node amount after esophagectomy <15 and ≥15 in clini-cal stage III esophageal squamous cell carcinoma (solid line: <15; dot line: ≥15).

Multiple-variable analysis was also performed using a Cox proportional hazards regression with Enter fashion. For overall survival, the pathologic tumor grade and number of lymph nodes had a significant impact. The *p*-values were 0.0162 and 0.0054 for pathologic tumor grade and lymph node amount, respectively. In recurrence-free patients, the pathologic tumor grade and lymph node amount contributed to a significant impact, with *p*-values of 0.0235 and 0.0124, respectively. The number of lymph nodes and pathological tumor grade significantly impacted disease-free survival. The *p*-values were 0.0385 and 0.0043 for the pathologic tumor grade and lymph nodes, respectively. All multivariable analysis data are summarized in Table 4.

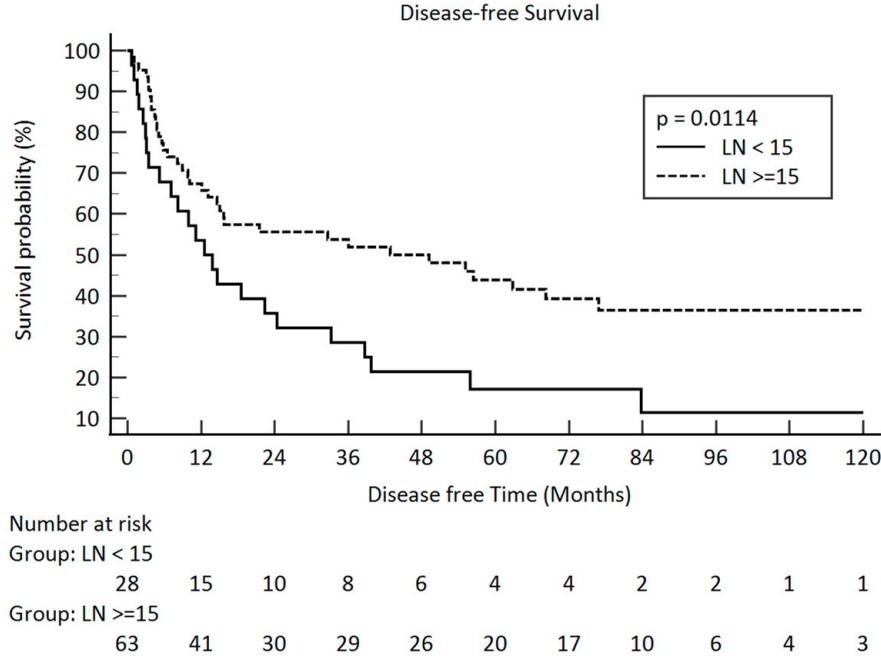

**Figure 3.** Disease-free survival in pathologic lymph node amount after esophagectomy <15 and ≥15 in clinical stage III esophageal squamous cell carcinoma (solid line: <15; dot line: ≥15).

**Table 4.** Results of multiple-variable analysis (Cox proportional hazards regression) of prognostic factors for overall survival (a) Free-To-Recurrence (b) and disease-free survival (c) in patients diagnosed with AJCC 8th clinical stage III esophageal cancer. (* statistically significant).

| (a) Overall Survival | | | | | |
|---|---|---|---|---|---|
| Covariate | b | Std. Error | Exp (b) | 95% CI of Exp (b) | *p*-Value |
| Clinical T stage | −0.1869 | 0.3067 | 0.8295 | 0.4547 to 1.5131 | 0.5422 |
| Clinical N stage | −0.1881 | 0.3023 | 0.8285 | 0.4581 to 1.4983 | 0.5337 |
| Primary Tumor Location | 0.1218 | 0.2103 | 1.1295 | 0.7480 to 1.7057 | 0.5624 |
| Pathologic Tumor Grade | 0.3762 | 0.1564 | 1.4567 | 1.0721 to 1.9794 | 0.0162 * |
| Lymph Node | −0.8304 | 0.2983 | 0.4359 | 0.2429 to 0.7822 | 0.0054 * |
| (b) Recurrence-Free Ratio | | | | | |
| Covariate | b | Std. Error | Exp (b) | 95% CI of Exp (b) | *p*-Value |
| Clinical T stage | −0.1980 | 0.4143 | 0.8203 | 0.3633 to 1.4218 | 0.6327 |
| Clinical N stage | −0.3303 | 0.348 | 0.7187 | 0.3633 to 1.4218 | 0.3427 |
| Primary Tumor Location | −0.0493 | 0.2425 | 0.9519 | 0.5918 to 1.5310 | 0.8388 |
| Pathologic Tumor Grade | 0.4222 | 0.1865 | 1.524 | 1.0584 to 2.1983 | 0.0235 * |
| Lymph Node | −0.8722 | 0.3487 | 0.418 | 0.2111 to 0.8279 | 0.0124 * |
| (c) Disease-Free Survival | | | | | |
| Covariate | b | Std. Error | Exp (b) | 95% CI of Exp (b) | *p*-Value |
| Clinical T stage | −0.2893 | 0.2854 | 0.7488 | 0.4280 to 1.3099 | 0.3106 |
| Clinical N stage | −0.0582 | 0.2812 | 0.9434 | 0.5437 to 1.6371 | 0.8359 |
| Primary Tumor Location | 0.04933 | 0.1968 | 1.0506 | 0.7143 to 1.5450 | 0.8021 |
| Pathologic Tumor Grade | 0.2851 | 0.1378 | 1.3299 | 1.0152 to 1.7422 | 0.0385 * |
| Lymph Node | −0.8114 | 0.2839 | 0.4442 | 0.2546 to 0.7750 | 0.0043 * |

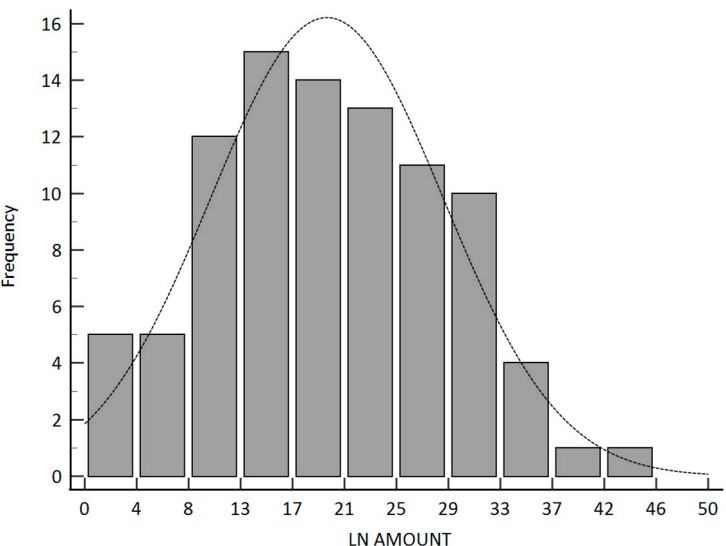

**Figure 4.** Contribution plot for the frequency of retrieved lymph node amount.

## 4. Discussion

This study analyzed the outcomes of different lymph node dissection amounts among patients with AJCC 8th-edition clinical stage III esophageal squamous cell carcinoma. All patients underwent concurrent chemoradiation therapy followed by esophagectomy with two-field lymph node dissections. Results revealed that the dissection amount had a significant impact on overall survival, even greater than the pathologic N stage. It also revealed the similar importance in "Free-To-Recurrence", indicating the time from the date of curative esophagectomy to the time of recurrence. Multiple-variable analysis with a Cox proportional hazards regression was employed to reduce the impact of different variables, highlighting the significance of surgical lymph node dissection. We also proofread the hypotheses.

Lymph node dissection can provide not only precise information about the current disease status, including the pathological N stage, but also curative impact. Lymph node dissection of <15 indicates that curative intent effect decreased, and some indicated that cancer may remain in the surgical field or the original cancer bed, which shortens the "Free-To-Recurrence" and worsens overall survival.

Patients undergoing concurrent chemoradiation followed by esophagectomy for radical lymphadenectomy have several issues. The first is the number of harvested lymph nodes affected by concurrent chemoradiation. Second, whether the lymph nodes are adequate and if this reduced amount could affect survival or recurrence remain concerning. In terms of the first issue, one hypothesis was that concurrent chemoradiation therapy leads to regression of lymph nodes [13]. In our study, we found that approximately 30% of patients with prior concurrent chemoradiation therapy could not obtain 15 lymph nodes after esophagectomy. This is consistent with the findings of other studies and hypotheses. Regarding the second issue, several studies have demonstrated similar conclusions. Jennifer et. al. found that survival significantly decreased in patients with <7 lymph nodes dissected for pathologic T3–4 patients. However, the impact of the number of dissected lymph nodes is not prominent in pathologic T1–2 [14]. However, most (78.9%) patients in the study by Jennifer et al. were diagnosed with esophageal adenocarcinoma, but not squamous cell carcinoma. In a study by Pamela et al., a large database review revealed that 10–15 dissected lymph nodes contributed to a better overall survival [15]. A large-scale database study by Wang et al. reported that 20 lymph node dissections improved overall survival and disease-specific survival in esophageal adenocarcinoma [16]. All of these studies support the similar perspective that the lymph node dissection amount will influence overall survival, which is consistent with our results. A literature review provided the benefits of radical lymphadenectomy among patients with esophageal cancer [7]. The

same conclusion about more lymphadenectomy providing better survival was recently published, both for squamous cell carcinoma and adenocarcinoma [8,9,17].

However, some studies have reported contradictory results. Jesper et al. conducted a retrospective study in a high-volume single center and revealed that the extent of lymphadenectomy does not influence survival after surgery for esophageal adenocarcinoma [10]. The hypothesis is that positive nodes indicate a disseminated disease, while non-metastatic nodes do not require removal; however, we think this hypothesis could not explain the higher survival result in our patients, even when all dissected lymph nodes were non-metastatic lesions. According to our study, only 19% (63 patients had >15 lymph nodes, and only 12 patients showed positive lymph nodes in the pathology report) of patients in the high-lymph-node group presented with positive lymph nodes in pathology. This indicates that 81% of patients had negative pathologic lymph node results, but still had higher survival rates, which was contrary to the hypothesis.

Regarding recurrence, the number of lymphadenectomies has been correlated with disease-free survival in some studies [16,17]; however, many studies have showed varying results [8,10]. In our series, lymph nodes dissection of <15 is a significant risk factor not only in Kaplan–Meier survival, but also in the Cox proportional hazards regression analysis in the variate of "Free-To-Recurrence".

This study aimed to eliminate confounding factors and limitations in patients with clinical stage III esophageal squamous cell carcinoma. All patients received the similar concurrent chemoradiation treatment protocol, and surgery was conducted by a highly experienced surgeon. Most of the currently published studies are in the mixed stage, increasing covariate interactions to lower the evidence. We provide real-world results for advanced stages to assess the significance of lymphadenectomy, along with information on administering adjuvant therapy after concurrent chemoradiation followed by esophagectomy. If lymphadenectomy cannot achieve an adequate level, the treatment team should consider adjuvant therapy to reduce the risk of recurrence.

Our study had several limitations. First, it was a retrospective study with a small number of patients; hence, many confounding factors affected the results due to the data design. Although we attempted to eliminate confounding factors such as excluding different pathologies and subjects lost to follow-up, the retrospective data series only provided limited evidence. Small sample size in both groups also reduced the statistical power, weakened the multivariance analysis and reduced the quality of this study. With the current sample size (n = 91), the statistical power is only 66%. We need to collect more patients to obtain 80% statistical power (total sample size should be more than 126 at least). Second, our series included patients from 1994 to 2023; the time period was over 30 years, potentially leading to many covariates not being predicted. We tried to shorten the time period from 2011 to 2019 and obtained the same result. Despite these limitations, we believe our findings are significant and worthy of publication. The data stem from real clinical practice, showing that esophagectomy remains a complex surgical procedure with high morbidity and mortality rates, even in high-volume medical centers. The limited number of cases is a challenge, necessitating a longer time span to gather more clinical data.

Many studies have faced similar challenges. For example, J. Peng, W. et al. (2005–2013) determined the minimal number of lymph nodes required for esophagectomy [18]. J. Lagergren, F. et al. (2000–2014) examined the impact of lymphadenectomy extent on outcomes [10]. C.M. Lo (2000–2015) explored how radiotherapy doses affect esophageal cancer treatment outcomes [19]. S. Sihag, T. et al. (1995–2017) found that extensive lymphadenectomy improves outcomes in advanced adenocarcinoma of the esophagus [17]. These studies have significantly contributed to current esophageal cancer treatment protocols even before they have been incorporated into long time span research.

According to published literature, lymphadenectomy after concurrent chemoradiotherapy lacks supporting evidence. Multiple-center randomized trials may improve the evidence level of this study and provide further evidence in the future.

## 5. Conclusions

In conclusion, a lymphadenectomy value of less than 15 is a high-risk factor for poor treatment outcomes among patients with AJCC 8th-edition clinical stage III esophageal squamous cell carcinoma receiving preoperative concurrent chemoradiation.

**Author Contributions:** Conceptualization: C.-M.L. and S.-H.L.; Data curation: S.-H.L. and C.-M.L.; Formal Analysis: C.-M.L.; Funding acquisition: H.-I.L. and S.-H.L.; Investigation: S.-T.L., Y.C. and L.-C.C.; Methodology: S.-T.L., Y.C. and L.-C.C.; Project administration: H.-I.L. and S.-H.L.; Resources: Y.-H.C. and S.-T.L.; Software: Y.C. and L.-C.C.; Supervision: H.-I.L., S.-H.L. and Y.-H.C.; Validation: Y.C. and L.-C.C.; Visualization: Y.C. and L.-C.C.; Writing—original draft: W.H.; Writing—review & editing: S.-H.L. and C.-M.L.; All authors have read and agreed to the published version of the manuscript.

**Funding:** This work was supported by grants from Chang Gung Memorial Hospital (CMRPG8M0903, CORPG8M0341, and CORPG8L1271).

**Institutional Review Board Statement:** The experimental protocol was established according to the ethical guidelines of the Declaration of Helsinki and approved by the Institutional Review Board of Chang Gung Memorial Hospital. In this retrospective design, the requirement for informed consent was waived, and the study protocol was approved by the Ethics Committee of Chang Gung Memorial Hospital (Institutional Review Board number: 202400100B0). IRB NO.: 202400100B0.

**Informed Consent Statement:** Not applicable.

**Data Availability Statement:** All data generated or analyzed during this study are included in this published article.

**Conflicts of Interest:** The authors declare no conflict of interest.

## Abbreviations

AJCC: American Joint Committee on Cancer.

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
