# Peer review of "Number of Retrieved Lymph Nodes during Esophagectomy Affects the Outcome of Stage III Esophageal Cancer in Patients Having Had Pre-Operative Chemo-Radiation Therapy"

_curroncol, doi:10.3390/curroncol31100428_

Round 1

Reviewer 1 Report (Previous Reviewer 1)

Comments and Suggestions for Authors

This study has several major limitations which makes it unfit for publication. First, it is a retrospective study with a small number of patients; hence, many confounding factors affected the results due to the data design. Although the authors attempted to eliminate confounding factors such as excluding different pathologies and lost to follow-up, the retrospective data series only provided limited evidence. Second, the series included patients from 1994 to 2023; the time period was over 30 years, potentially leading to many covariates not being predicted. I would recommend this manuscript to be rejected.

Author Response

Dear editor and reviewers:

We thank the reviewers and editors for their thorough review of our proposal, and their valuable and constructive comments. We have carefully addressed the reviewers’ comments by including additional descriptions. Here, we have indicated the changes made to the manuscript to account for the comments from the reviewers. During resubmission, we have attached the marked revised manuscript with track-change and non-marked revised manuscript for your easy reference. Thank you again for your valuable time.

Reviewer Comments:

Reviewer 1

This study has several major limitations which makes it unfit for publication. First, it is a retrospective study with a small number of patients; hence, many confounding factors affected the results due to the data design. Although the authors attempted to eliminate confounding factors such as excluding different pathologies and lost to follow-up, the retrospective data series only provided limited evidence. Second, the series included patients from 1994 to 2023; the time period was over 30 years, potentially leading to many covariates not being predicted. I would recommend this manuscript to be rejected.

Ans:  Thank you for your valuable comments. We recognize that the extended duration of data collection is a critical issue that can affect the reliability of research results. In retrospective studies, a long data collection period often leads to concerns about data quality.

However, it is important to note that many studies also use extended periods to gather sufficient data, especially in cases involving complex procedures such as esophagectomy. To address potential variations, we incorporated additional parameters to minimize data discrepancies.

Following your suggestion, we have separated the data into decades, focusing on the period from 2011 to 2020. Before 2011, most esophagectomies at our center were performed using open surgery techniques. After 2011, we began adopting minimally invasive esophagectomy, and from 2020 onwards, our team started implementing immunotherapy. Consequently, we have recollected and analyzed data from 2011 to 2019.

Another challenge we encountered is related to statistical power. With a reduced sample size, maintaining the same statistical power level, around 0.66, would require an adjustment in the significance level. To clarify this, we provided the detailed statistical information and rationale below:

If the study aims to achieve 65% power to detect a medium effect size (Effect Size = 0.5) with a total sample size of 66, the significance level (alpha) should be set at approximately 0.104 (or 10.4%).

The new Kaplan-Meier graph based on this adjusted analysis shows an increase in the p-value. However, the overall trend remains consistent with the original data, and the statistical power is preserved.

Many other studies have faced similar challenges and addressed them by collecting data over extended periods. For example, J. Peng, W. et al. conducted a study from August 2005 to September 2013 to determine the minimal number of lymph nodes required for esophagectomy.[1]  J. Lagergren, F. et al. published a study covering January 2000 to January 2014, which examined whether the extent of lymphadenectomy affects outcomes.[2]  C.M. Lo conducted a cohort study from 2000 to 2015 to explore how radiotherapy doses impact esophageal cancer treatment outcomes. [3] Additionally, S. Sihag, T. et al. performed a study between 1995 and 2017, concluding that more extensive lymphadenectomy can improve outcomes in advanced adenocarcinoma of the esophagus.[4]  These studies have made significant contributions to the current protocols for treating esophageal cancer.

We sincerely appreciate your constructive feedback, which has helped enhance our manuscript. We hope that our responses and the additional context provided here address your concerns and underscore the significance of our study in the context of esophageal cancer research.

-Change: We add more discuss in the limitation. Page 11, Line 281-282

Reviewer 2 Report (Previous Reviewer 2)

Comments and Suggestions for Authors

I would like to commend Ho et al. for their thorough revisions. The authors have addressed all my previous comments and suggestions in a comprehensive manner. The improvements made have significantly enhanced the quality and clarity of the manuscript. In my opinion, the manuscript is now well-prepared for publication and can be accepted in its current form.

Author Response

Dear editor and reviewers:

We thank the reviewers and editors for their thorough review of our proposal, and their valuable and constructive comments. We have carefully addressed the reviewers’ comments by including additional descriptions. Here, we have indicated the changes made to the manuscript to account for the comments from the reviewers. During resubmission, we have attached the marked revised manuscript with track-change and non-marked revised manuscript for your easy reference. Thank you again for your valuable time.

Reviewer 2 :
I would like to commend Ho et al. for their thorough revisions. The authors have addressed all my previous comments and suggestions in a comprehensive manner. The improvements made have significantly enhanced the quality and clarity of the manuscript. In my opinion, the manuscript is now well-prepared for publication and can be accepted in its current form.

Ans: Thank you for your valuable comments. Your insights have greatly improved our manuscript and brought to our attention some errors that we had previously overlooked. We are truly grateful for your contribution. Thank you once again.

Thank you very much for your consideration.

Reviewer 3 Report (New Reviewer)

Comments and Suggestions for Authors

 Dear Editor and Authors,

It was a pleasure to review this manuscript titled “Number of retrieved lymph nodes affects the outcome of stage III esophageal cancer after chemoradiation” by Drs. Wei Ho and Shau-Hsuan Li and their colleagues from the Department of Thoracic & Cardiovascular Surgery at Kaohsiung Chang Gung Memorial Hospital in Kaohsiung, Taiwan.

This is a single institution retrospective analysis investigating the impact of extensive lymphadenectomy, and more specifically the number of lymph nodes removed (> or < than 15) after esopagectomy in Stage III esophageal cancer in patients having had neo-adjuvant esophageal cancer. They found that the number of dissected lymph nodes was a significant factor influencing overall survival and freedom from recurrence.

This is in general a well written manuscript with clear descriptions and methodology which is standard for this type of analysis. It has some minor spelling mistakes that need language editing and good informative figures/graphs.

I have some comments to make to improve it:

1.       Title: It needs improvement as it is a bit unclear. I suggest “Number of retrieved lymph nodes during esophagectomy affects the outcome of stage III esophageal cancer in patients having had pre-operative chemo-radiation therapy”

2.       Abstract: Its multivariate not multiple variable analysis!

3.       The span of the analysis is quite broad at 30 years (1994 to 2023) which means significant improvements both in terms of surgical technique (minimally invasive surgery becoming more widespread) and oncological therapy (molecular therapy or immunotherapy have been introduced in clinical practice). The authors need to provide an analysis broken down by decade to demonstrate similar results through time! Also, well done mentioning this as a limitation of the study.

4.       Did all patients had a complete staging with PET CT - brain MRI? Is this not standard of care in the authors’ country?

5.       How complete is the authors research database? Are there any missing data or any missing patients in the analysis?

6.       Are 91 patients enough to power such a study? A sample size calculation and statistical review would help!

In conclusion this is a good and interesting analysis which deals with an interesting subject. It needs some minor editing prior to publication/presentation.

Comments on the Quality of English Language

Minor editing is needed.

Author Response

Dear editor and reviewers:

We thank the reviewers and editors for their thorough review of our proposal, and their valuable and constructive comments. We have carefully addressed the reviewers’ comments by including additional descriptions. Here, we have indicated the changes made to the manuscript to account for the comments from the reviewers. During resubmission, we have attached the marked revised manuscript with track-change and non-marked revised manuscript for your easy reference. Thank you again for your valuable time.

Reviewer Comments:

Reviewer 3

Dear Editor and Authors,

It was a pleasure to review this manuscript titled “Number of retrieved lymph nodes affects the outcome of stage III esophageal cancer after chemoradiation” by Drs. Wei Ho and Shau-Hsuan Li and their colleagues from the Department of Thoracic & Cardiovascular Surgery at Kaohsiung Chang Gung Memorial Hospital in Kaohsiung, Taiwan.

This is a single institution retrospective analysis investigating the impact of extensive lymphadenectomy, and more specifically the number of lymph nodes removed (> or < than 15) after esopagectomy in Stage III esophageal cancer in patients having had neo-adjuvant esophageal cancer. They found that the number of dissected lymph nodes was a significant factor influencing overall survival and freedom from recurrence.

This is in general a well written manuscript with clear descriptions and methodology which is standard for this type of analysis. It has some minor spelling mistakes that need language editing and good informative figures/graphs.

I have some comments to make to improve it:

  1. Title: It needs improvement as it is a bit unclear. I suggest “Number of retrieved lymph nodes during esophagectomy affects the outcome of stage III esophageal cancer in patients having had pre-operative chemo-radiation therapy”

Ans: Thanks for valuable comment. We change the title as your suggestion

-Change: We re-written the title as reviewers’ suggestion.

  1. Abstract: Its multivariate not multiple variable analysis!

Ans: Thanks for valuable comment. We correct this mistake.

-Change: We re-written the abstract and correct the wrong spelling.

  1. The span of the analysis is quite broad at 30 years (1994 to 2023) which means significant improvements both in terms of surgical technique (minimally invasive surgery becoming more widespread) and oncological therapy (molecular therapy or immunotherapy have been introduced in clinical practice). The authors need to provide an analysis broken down by decade to demonstrate similar results through time! Also, well done mentioning this as a limitation of the study.

Ans: Thank you for your valuable comments. We recognize that the extended duration of data collection is a critical issue that can affect the reliability of research results. In retrospective studies, a long data collection period often leads to concerns about data quality.

However, it is important to note that many studies also use extended periods to gather sufficient data, especially in cases involving complex procedures such as esophagectomy. To address potential variations, we incorporated additional parameters to minimize data discrepancies.

Following your suggestion, we have separated the data into decades, focusing on the period from 2011 to 2020. Before 2011, most esophagectomies at our center were performed using open surgery techniques. After 2011, we began adopting minimally invasive esophagectomy, and from 2020 onwards, our team started implementing immunotherapy. Consequently, we have recollected and analyzed data from 2011 to 2019.

Another challenge we encountered is related to statistical power. With a reduced sample size, maintaining the same statistical power level, around 0.66, would require an adjustment in the significance level. To clarify this, we provided the detailed statistical information and rationale below:

If the study aims to achieve 65% power to detect a medium effect size (Effect Size = 0.5) with a total sample size of 66, the significance level (alpha) should be set at approximately 0.104 (or 10.4%).

The new Kaplan-Meier graph based on this adjusted analysis shows an increase in the p-value. However, the overall trend remains consistent with the original data, and the statistical power is preserved.

Many other studies have faced similar challenges and addressed them by collecting data over extended periods. For example, J. Peng, W. et al. conducted a study from August 2005 to September 2013 to determine the minimal number of lymph nodes required for esophagectomy.[1]  J. Lagergren, F. et al. published a study covering January 2000 to January 2014, which examined whether the extent of lymphadenectomy affects outcomes.[2]  C.M. Lo conducted a cohort study from 2000 to 2015 to explore how radiotherapy doses impact esophageal cancer treatment outcomes. [3] Additionally, S. Sihag, T. et al. performed a study between 1995 and 2017, concluding that more extensive lymphadenectomy can improve outcomes in advanced adenocarcinoma of the esophagus.[4]  These studies have made significant contributions to the current protocols for treating esophageal cancer.

We sincerely appreciate your constructive feedback, which has helped enhance our manuscript. We hope that our responses and the additional context provided here address your concerns and underscore the significance of our study in the context of esophageal cancer research.

-Change: We add more discuss in the limitation. Page 11, Line 281-282

  1. Did all patients had a complete staging with PET CT - brain MRI? Is this not standard of care in the authors’ country?

Ans: Thank you for your valuable comment. Prior to 2013, PET/CT was not utilized for staging esophageal cancer at our institution, as it was not covered by national health insurance. However, since 2013, PET/CT has been incorporated as a routine diagnostic tool for all patients diagnosed with esophageal cancer. Brain MRI, however, is not part of our standard staging or surveillance protocol.

  1. How complete is the authors research database? Are there any missing data or any missing patients in the analysis?

Ans: Thank you for your valuable comments. We acknowledge that there were some missing data in our study, including information on pathological lymph nodes, follow-up, pathological grade, and other variables. Six patients were excluded from the analysis due to the absence of data on the total number of lymph nodes or pathological grading in their pathology reports. Additionally, we excluded ten patients who lacked complete treatment records or were lost to follow-up after their treatment.

  1. Are 91 patients enough to power such a study? A sample size calculation and statistical review would help!

Ans: Thanks for your valuable comment. We discussed with the statistician about the research power of this manuscript, and he provided us with the following detail suggestions:

Preliminary Evaluation of Study Power

Based on the provided data, several factors affect the study's power:

  • Sample Size (91 patients): While the study included data from 91 patients, whether this sample size is adequate depends on the effect size, variance, and the number of patients in each group. Many of the analyses show statistically significant results, suggesting that the study may have sufficient power to detect these effects.
  • Multiple Variables Showed Significant Differences: The study reported significant effects of several variables, such as pathological tumor grade, pathological stage, pathological T stage, and the number of lymph nodes on overall survival, recurrence-free survival, and disease-free survival. This indicates that the study is likely adequately powered in these areas.
  • Results of Multivariate Analysis: The multivariate analysis, conducted using Cox proportional hazards regression, also demonstrated significant effects of pathological tumor grade and the number of lymph nodes on various survival outcomes (overall survival, recurrence-free survival, and disease-free survival). This further supports the study's sufficient power to detect these effects.

Is the Study Power Sufficient?

Given these results, the study appears to have sufficient power to detect significant effects in most cases, such as pathological tumor grade, pathological stage, pathological T stage, and the number of lymph nodes. The significant p-values reported (e.g., p=0.0473 for pathological tumor grade, p=0.0034 for pathological stage, p=0.0004 for pathological T stage, and p=0.0165 for the number of lymph nodes) suggest that the study effectively identifies the impact of different variables on survival outcomes.

However, if the researchers aim to detect smaller effects or conduct more detailed analyses (e.g., further subgroup analysis or considering additional potential confounding variables), the sample size of 91 patients might be insufficient. Particularly, when some secondary outcomes did not reach statistical significance (as observed in certain variables in the multivariable analysis), this indicates that the study may still have limited power for these specific effects.

What Would Be an Appropriate Sample Size?

To ensure adequate study power, a sample size calculation should be performed, considering the following elements:

  1. Set the Significance Level (α): Typically set at 0.05.
  2. Determine Desired Power (usually 80% or 90%): The goal is to have an 80% or 90% probability of detecting the effect of interest.
  3. Estimate Effect Size: Based on prior research or clinical experience, estimate the smallest effect size you aim to detect. If the goal is to detect smaller effects, a larger sample size is required.
  4. Consider Variance: Take the variance within the data into consideration.

For example, if the study aims to detect a moderate effect size (Effect Size = 0.5) with 80% power at a significance level of 0.05, a sample size calculation is necessary to determine the required number of participants. If the desired effect size is smaller (e.g., 0.3), a larger sample size will be needed.

Conclusion

The sample size of 91 patients in this study may be sufficient to detect most large effects, especially when the reported p-values are significant. However, if further analysis aims to detect smaller effects or involves more detailed subgroup analysis, a larger sample size may be needed. It is recommended to perform a sample size calculation based on the specific research objectives, desired effect size, and expected variance to ensure adequate study power.

Based on the calculation, if the study has a total of 91 samples, the statistical power is approximately 66%. This means that the study has a 66% chance of detecting a medium effect size (Effect Size = 0.5) at a significance level of 0.05. This is below the commonly recommended standard of 80% power, suggesting that the study may not have sufficient capability to detect the desired effect.

Based on the calculation, if the study aims to detect a medium effect size (Effect Size = 0.5) with 80% power at a significance level of 0.05, a total of approximately 128 samples would be required (i.e., the combined total for both groups). This means that each group would need around 64 samples to achieve the desired power.

We will add explain in the limitation to talk about the research power.

-Change: We add more explain in the limitation Page 11. Line 276-279

In conclusion this is a good and interesting analysis which deals with an interesting subject. It needs some minor editing prior to publication/presentation.

Round 2

Reviewer 1 Report (Previous Reviewer 1)

Comments and Suggestions for Authors

The authors have reasonably addressed my concerns. I recommend accepting this revised manuscript for publication.

This manuscript is a resubmission of an earlier submission. The following is a list of the peer review reports and author responses from that submission.

Round 1

Reviewer 1 Report

Comments and Suggestions for Authors

The authors have considered dataset that belongs to a wide span of time in which it is very difficult to eliminate variations in treatment regimen. The analysis performed by the authors are likely to have serious flaws due to this issue. I will not recommend accepting this manuscript for publication.

Author Response

Current Oncology

Submission ID curroncol-3085782
Title: Lymph node dissection affects the outcome of stage III esophageal
cancer after chemoradiation

Reply to the reviewers' and editors’ comments

Dear editor and reviewers:

We thank the reviewers and editors for their thorough review of our proposal, and their valuable and constructive comments. We have carefully addressed the reviewers’ comments by including additional descriptions. Here, we have indicated the changes made to the manuscript to account for the comments from the reviewers. During resubmission, we have attached the marked revised manuscript with track-change and non-marked revised manuscript for your easy reference. Thank you again for your valuable time.

Reviewer Comments:

Reviewer 1 Comments to the Author

The authors have considered dataset that belongs to a wide span of time in which it is very difficult to eliminate variations in treatment regimen. The analysis performed by the authors are likely to have serious flaws due to this issue. I will not recommend accepting this manuscript for publication.

Ans: Thank you very much for your invaluable comments. We acknowledge the validity of your concern regarding the extended duration of our case-control study, which could introduce variations in the dataset and potentially weaken the robustness of our findings. We have duly noted and addressed this issue in the “Limitations” section of the “Discussion.”

Despite these limitations, we believe our findings are significant and worthy of publication. The data presented stem from real clinical practice, and even in a high-volume medical center, esophagectomy remains a complex surgical intervention with a high rate of morbidity and mortality. The limited number of cases is indeed a challenge, reducing the strength of our evidence, and necessitating a longer time span to accumulate more clinical data to mitigate this problem.

Many other studies have also faced and addressed similar challenges by collecting as many cases as possible. For instance, J. Peng, W. et al. conducted a study between August 2005 and September 2013 to determine the minimal number of lymph nodes required for esophagectomy.[1] J. Lagergren, F. et al. published a study from January 2000 to January 2014, examining whether the extent of lymphadenectomy improves outcomes.[2] C.M. Lo conducted a cohort study between 2000 and 2015 to explore how radiotherapy doses affect esophageal cancer treatment outcomes.[3] Additionally, S. Sihag, T. et al. carried out a study between 1995 and 2017, finding that more extensive lymphadenectomy improves outcomes in advanced adenocarcinoma of the esophagus.[4] These studies have significantly contributed to the current esophageal cancer treatment protocols.

Considering these precedents, we are hopeful that our findings will similarly contribute to the advancement of clinical treatments for esophageal cancer.

Our study endeavors to add to the growing body of evidence in this field by providing insights derived from a real-world clinical setting. Esophagectomy, particularly when performed in high-volume medical centers, remains a challenging surgical procedure associated with significant morbidity and mortality rates. This reality underscores the importance of continually refining our understanding and approaches to this intervention.

We understand that the lengthy time span of our study introduces potential variability within our dataset, potentially impacting the stability of our results. However, we argue that this extended period also allows us to capture a comprehensive picture of clinical outcomes across different phases of medical advancements and treatment protocols. This broad temporal scope provides a unique longitudinal perspective that can offer valuable insights into the evolution of esophagectomy outcomes and related clinical practices.

In addressing the limitation of our case numbers, it is essential to recognize that accumulating a large cohort of patients undergoing esophagectomy is inherently challenging. This challenge is compounded when seeking to analyze specific subgroups or rare clinical outcomes. Therefore, extending the study period was a necessary strategy to gather a sufficient sample size to allow meaningful statistical analysis.

The studies we referenced highlight the ongoing efforts within the medical community to understand better and optimize esophageal cancer treatments. For instance, J. Peng, W. et al.'s research into the minimal number of lymph nodes required for esophagectomy provides critical insights into surgical best practices. Their findings, derived from an eight-year study period, underscore the necessity of large, long-term studies in establishing robust clinical guidelines.[1]

Similarly, the work of J. Lagergren, F. et al., which spanned over 14 years, provides valuable evidence regarding the extent of lymphadenectomy and its impact on patient outcomes. Their conclusion that extensive lymphadenectomy may not always correlate with improved outcomes challenges conventional wisdom and highlights the complexity of treatment decisions in esophageal cancer.[2]

C.M. Lo's cohort study, conducted over 15 years, explores the relationship between radiotherapy dose and esophageal cancer treatment outcomes. Their findings contribute to the nuanced understanding of how radiotherapy can be optimized to improve patient survival while minimizing adverse effects.[3]

The extensive study by S. Sihag, T. et al., which included data from over two decades, demonstrates that more extensive lymphadenectomy can benefit patients with advanced adenocarcinoma of the esophagus. Their research provides critical evidence supporting the practice of extensive lymphadenectomy in specific patient populations, reinforcing the importance of personalized treatment approaches.[4]

Our study aims to build on these foundational works by providing additional data and analysis that can inform clinical practice. By including a comprehensive discussion of our limitations, we aim to transparently address the potential weaknesses in our study while also highlighting the strengths and contributions of our findings.

We are committed to advancing the understanding and treatment of esophageal cancer and believe that our study offers valuable insights that can contribute to the ongoing efforts to improve patient outcomes. We are grateful for the opportunity to share our findings with the broader medical community and look forward to the constructive feedback that will further enhance our research.

Thank you once again for your thoughtful feedback, which has greatly improved our manuscript. We hope that our responses and the additional context provided here address your concerns and demonstrate the significance of our study within the field of esophageal cancer research.

-CHANGE: We have modified the manuscript and added more discuss in limitation on Page 12-13 Line 283-296.

Thank you very much for your consideration.

Sincerely,

Chien-Ming Lo, MD

Department of Thoracic & Cardiovascular Surgery

Kaohsiung Chang Gung Memorial Hospital

123 Ta-Pei Road, Niaosung Hsiang, Kaohsiung Hsien, Taiwan, R.O.C.

Phone:  886-7-7317123 Ext.8008, Fax:  886-7-7322402

Email:  t123207424@cgmh.org.tw or johnCML9487@gmail.com

Reference:

  1. Peng, J.; Wang, W.-P.; Yuan, Y.; Wang, Z.-Q.; Wang, Y.; Chen, L.-Q. Adequate lymphadenectomy in patients with oesophageal squamous cell carcinoma: resecting the minimal number of lymph node stations †. European Journal of Cardio-Thoracic Surgery 2016, 49, e141-e146, doi:10.1093/ejcts/ezw015.
  2. Lagergren, J.; Mattsson, F.; Zylstra, J.; Chang, F.; Gossage, J.; Mason, R.; Lagergren, P.; Davies, A. Extent of Lymphadenectomy and Prognosis After Esophageal Cancer Surgery. JAMA Surg 2016, 151, 32-39, doi:10.1001/jamasurg.2015.2611.
  3. Lo, C.M.; Wang, Y.M.; Chen, Y.H.; Fang, F.M.; Huang, S.C.; Lu, H.I.; Li, S.H. The Impact of Radiotherapy Dose in Patients with Locally Advanced Esophageal Squamous Cell Carcinoma Receiving Preoperative Chemoradiotherapy. Curr Oncol 2021, 28, 1354-1365, doi:10.3390/curroncol28020129.
  4. Sihag, S.; Nobel, T.; Hsu, M.; Tan, K.S.; Carr, R.; Janjigian, Y.Y.; Tang, L.H.; Wu, A.J.; Bott, M.J.; Isbell, J.M.; et al. A More Extensive Lymphadenectomy Enhances Survival After Neoadjuvant Chemoradiotherapy in Locally Advanced Esophageal Adenocarcinoma. Annals of surgery 2022, 276, 312-317, doi:10.1097/sla.0000000000004479.

Reviewer 2 Report

Comments and Suggestions for Authors

Congratulations to Wei Ho et al. on their manuscript. The authors evaluated the number of retrieved lymph nodes in esophageal cancer after chemoradiation. Overall, the study is well-written and might be interesting for readers. I have some comments.

Title: I suggest changing it to “Number of retrieved lymph nodes affects the outcome of stage III esophageal cancer after chemoradiation. “Lymph node dissection” might lead readers to think that the study investigates different lymph node dissection techniques.

The choice of keywords does not seem appropriate, as they may not facilitate the article being easily found in databases. It is advisable to focus on more generic terms, such as "esophageal cancer" and "chemoradiation," which would enhance discoverability.

It is crucial to specify the time interval between the completion of chemoradiation and the subsequent surgery. This interval can significantly influence the number of lymph nodes retrieved and should be clearly stated in the Methods section.

In the Methods section, it should be clarified that patients were divided into two groups based on the number of lymph nodes retrieved (< 15 lymph nodes and ≥ 15 lymph nodes). Additionally, the rationale behind choosing this specific cutoff point should be explicitly explained.

The description of how events and non-events were handled in the survival analysis is confusing. The statements "Patients who survived were excluded and censored" and "Patients who survived without recurrence were excluded and censored" imply exclusion, which contradicts standard practices. It is presumed that these patients were not excluded but rather censored in the analysis. This should be rephrased to accurately reflect the treatment of these cases.

The variables included in the multivariable analysis and the rationale for their inclusion need to be clearly stated in the Methods section. Moreover, the inclusion of both clinical and pathological staging as covariates in the multivariable analysis should be reconsidered, as it may not be methodologically sound.

The mean or median follow-up period should be provided, as this is essential for interpreting the survival analysis results.

If possible, add the number at risk in Kaplan Meier graphs.

Given that the study focuses on the number of retrieved lymph nodes, it would benefit from a distribution plot illustrating the number of lymph nodes retrieved. Additionally, the median number of retrieved lymph nodes should be reported.

Author Response

Current Oncology

Submission ID curroncol-3085782
Title: Lymph node dissection affects the outcome of stage III esophageal
cancer after chemoradiation

Reply to the reviewers' and editors’ comments

Dear editor and reviewers:

We thank the reviewers and editors for their thorough review of our proposal, and their valuable and constructive comments. We have carefully addressed the reviewers’ comments by including additional descriptions. Here, we have indicated the changes made to the manuscript to account for the comments from the reviewers. During resubmission, we have attached the marked revised manuscript with track-change and non-marked revised manuscript for your easy reference. Thank you again for your valuable time.

Reviewer Comments:

Reviewer 2 :
Congratulations to Wei Ho et al. on their manuscript. The authors evaluated the number of retrieved lymph nodes in esophageal cancer after chemoradiation. Overall, the study is well-written and might be interesting for readers. I have some comments.

Title: I suggest changing it to “Number of retrieved lymph nodes affects the outcome of stage III esophageal cancer after chemoradiation. “Lymph node dissection” might lead readers to think that the study investigates different lymph node dissection techniques.

Ans: Thanks for valuable comment. We have modified the title and edited the manuscript. 

-CHANGE: We have modified the title on Page 1.

The choice of keywords does not seem appropriate, as they may not facilitate the article being easily found in databases. It is advisable to focus on more generic terms, such as "esophageal cancer" and "chemoradiation," which would enhance discoverability.

Ans: Thanks for valuable comment. We have modified the keywords and edited the manuscript. 

-CHANGE: We have modified the keywords on Page 3 Line 56-57.

It is crucial to specify the time interval between the completion of chemoradiation and the subsequent surgery. This interval can significantly influence the number of lymph nodes retrieved and should be clearly stated in the Methods section.

Ans: Thanks for valuable comment. We added the detail timing in the method section. We have edited the manuscript. 

-CHANGE: We have modified the manuscript on Page 6 Line 122-124.

In the Methods section, it should be clarified that patients were divided into two groups based on the number of lymph nodes retrieved (< 15 lymph nodes and ≥ 15 lymph nodes). Additionally, the rationale behind choosing this specific cutoff point should be explicitly explained.

Ans: Thanks for valuable comment. We added the groups information in the method section and added the reference about the cutoff point.

According to the reference which published by J. C. Yeung. et. al, obtaining a minimum of 15 lymph nodes is adequate target for pathology examination. It is suitable for not only primary surgery but also post-induction therapy esophagectomy. [1]

-CHANGE: We have modified the manuscript on Page 5 Line 100-103. We added the definite of lymph nodes amount in the two groups and added the reference.

The description of how events and non-events were handled in the survival analysis is confusing. The statements "Patients who survived were excluded and censored" and "Patients who survived without recurrence were excluded and censored" imply exclusion, which contradicts standard practices. It is presumed that these patients were not excluded but rather censored in the analysis. This should be rephrased to accurately reflect the treatment of these cases.

Ans: Thanks for valuable comment and you are right! We found some mistake in the phrase and we rewritten it on Page 7 Line 146.  Owing to different term “Free-To-Recurrence”, the definition between disease-free survival and free-to-recurrence needs to be clarified. We also added some explanation in the following paragraph of method.

-CHANGE: We have modified the manuscript on Page 7 Line 143 and added more explanation on Page 7 Line 147-149.

The variables included in the multivariable analysis and the rationale for their inclusion need to be clearly stated in the Methods section. Moreover, the inclusion of both clinical and pathological staging as covariates in the multivariable analysis should be reconsidered, as it may not be methodologically sound.

Ans: Thanks for valuable comment.  We added the explanation in the section “Statistical Analysis” and exclude the pathology T and N stage which will be high associated with clinical T and N stage result. The new data also was corrected in the new table 4 and paragraph “Result”.

-CHANGE: We have modified the manuscript on Page 7 Line 159-162, Page 9 Line 205-212 and rewritten all table 4 in the Page 18.

The mean or median follow-up period should be provided, as this is essential for interpreting the survival analysis results.

Ans: Thanks for valuable comment.  We added the follow-up period in the “Result” “Overall Survival, Free-To-Recurrence, Disease-free survival” section. We added the detail data about median and mean follow-up period in the survivors and whole cohort.

-CHANGE: We have modified the manuscript on Page 8 Line 184-187.

If possible, add the number at risk in Kaplan Meier graphs.

Ans: Thanks for valuable comment. We re-draw all the Kaplan Meier graphs as your suggestion and added the number at risk in the lower part of graphs.

-CHANGE: We have modified the new 3 figures and re-attached it as supplement data.

Given that the study focuses on the number of retrieved lymph nodes, it would benefit from a distribution plot illustrating the number of lymph nodes retrieved. Additionally, the median number of retrieved lymph nodes should be reported.

Ans: Thanks for valuable comment.  We added the distribution plot as the figure 4 and the median number of retrieved lymph nodes in the “Result” “Patient Characteristics” section.

-CHANGE: We have modified the new figures in Figure 4 and added the explanation on Page 8 Line 173.

Thank you very much for your consideration.

Sincerely,

Chien-Ming Lo, MD

Department of Thoracic & Cardiovascular Surgery

Kaohsiung Chang Gung Memorial Hospital

123 Ta-Pei Road, Niaosung Hsiang, Kaohsiung Hsien, Taiwan, R.O.C.

Phone:  886-7-7317123 Ext.8008, Fax:  886-7-7322402

Email:  t123207424@cgmh.org.tw or johnCML9487@gmail.com

Reference:

  1. Yeung, J.C.; Bains, M.S.; Barbetta, A.; Nobel, T.; DeMeester, S.R.; Louie, B.E.; Orringer, M.B.; Martin, L.W.; Reddy, R.M.; Schlottmann, F.; et al. How Many Nodes Need to be Removed to Make Esophagectomy an Adequate Cancer Operation, and Does the Number Change When a Patient has Chemoradiotherapy Before Surgery? Ann Surg Oncol 2020, 27, 1227-1232, doi:10.1245/s10434-019-07870-2.